# Understanding Farm-Level Incentives within the Bioeconomy Framework: Prices, Product Quality, Losses, and Bio-Based Alternatives

**Sarah Jansen** [1,*]**, William Foster** [1,2]**, Gustavo Anríquez** [1,2] **and Jorge Ortega** [1]

1   Departamento de Economía Agraria, Facultad de Agronomía e Ingeniería Forestal, Pontificia Universidad Católica de Chile, 7820436 Santiago, Chile; wfoster@uc.cl (W.F.); gustavo.anriquez@uc.cl (G.A.); jortegao@uc.cl (J.O.)
2   Millennium Nucleus Center for the Socioeconomic Impact of Environmental Policies (CESIEP), 7820436 Santiago, Chile
*   Correspondence: sjansen1@uc.cl

**Abstract:** The bioeconomy framework emphasizes potential contributions of life sciences to novel, bio-based products and to discover economic uses for what would otherwise be considered waste or loss in traditional production systems. To best exploit this perspective, especially for biowaste innovations, economists should develop behavioral models that integrate decision-making with biophysical concepts. The supply to bioeconomy uses of farm production otherwise lost depends on the relative net benefits of adjusting production across a range of quality levels. Without understanding such incentives, one cannot fully anticipate the effects on prices and consumer welfare due to new alternatives. The analysis here examines farm-level incentives that determine quality, sales and loss levels, and possible switching of supplies to alternative uses. We present a farmer decision model of the distribution of product qualities, total losses, and the adoption of alternative profitable activities, such as for antioxidants or other novel bioproducts. We demonstrate how the introduction of bio-based alternatives changes opportunity costs of resource use, altering product quality proportions and sales to traditional markets. Adopting biowaste alternatives depends on scale, productivity, and fixed costs; adopting these reduces the proportion of production going to traditional buyers/consumers and shifts downward the distribution of traditional product (e.g., food-grade) qualities.

**Keywords:** bioeconomy policy; farm product waste and losses; food quality; farmer behavior; technology adoption

## 1. Introduction

World population increases will shift the total demand for food, fuel, and raw materials, putting pressure on natural resource availability and altering patterns of production and consumption. In response, various institutions stress the promotion of alternative production techniques in global food and agricultural production systems [1,2]. Bioeconomy strategies have been widely considered in national plans for research and development, not only as potential innovations and a contribution to economic growth, but also as a strategy to reduce dependency on non-renewable resources [3]. Broadly speaking, the bioeconomic approach to farm production may be understood as contributing to the shift of economic systems from traditional uses of non-renewable resources to alternative uses of renewable resources. A key factor in promoting this shift would be the adoption, at the level of individual decision makers, of production strategies derived from advances in the life sciences to produce non-traditional goods and services [4,5]. While farmers have traditionally been suppliers of primary products (most importantly food), as the bioeconomy develops, farmers will participate in the production of primary materials for novel, non-traditional products and in the management of sustainable agricultural systems [6].

The introduction of bioeconomy-based products adds to farmers' portfolios of potential income sources and an increase in the economic efficiency of resource use [7]. In particular, biomass production that would otherwise go unsold in traditional contexts, and which would be described as a waste or loss, might be profitably used as alternative products destined for new markets with associated prices as incentives [8,9]. This link between the introduction of markets for new, bioeconomy-derived goods and agricultural product waste or losses has yet to be well explored at the farm decision level and is the focus of this present paper. It involves questions related to control over resources that will alter the volume and quality distribution of production. Policies aimed at the promotion of bioeconomy alternatives will have implications for other markets by shifting the use of these resources away from a traditional, final-consumer orientation, perhaps leading to unintended consequences with unanticipated net benefits. A review of the scientific literature concerned with the sustainability of the bioeconomy concludes that, "the bioeconomy cannot be considered self-evidently sustainable" [10] (p. 243). The tension between the use of resources aimed at traditional markets and their use for new bio-based alternatives presents a challenge to analysts to better understand how producers at the micro-level make decisions and the implications for the balance between affordable food and expanding the range of nonfood, biomass products [5].

The competing claims on biomass for traditional uses and for securing a sustainable biomass supply for a future, mature bioeconomy will require an understanding of the drivers of competition over resource use [11]. Methodological approaches to examining aggregate farmer responses to the introduction of bioeconomy alternatives have focused on the description of the potential competition for land used traditionally for biomass for food and fiber [12]. Authors have examined, via simulations, how farmers react in the aggregate in order to project the impacts of bio-based products (the literature has focused on biofuels) on the future use of land for food and the implications for greenhouse gas emissions (e.g., [13,14]). This present paper makes a contribution to this understanding by extending the methodological framework regarding resource decisions in a bioeconomic context, which has emphasized the competition in land allocation between distinct products, to the question of how individual farmers make decisions over the distribution of the quality characteristics of their biomass production. Our modeling approach is directly applicable for data at the scale of farms, but has implications for aggregate, market-level outcomes, such as quantities supplied across a range of traditional product qualities and rates of participation of farmers in bioeconomy activities.

The introduction of alternative, bio-based products will also have consequences for the use of what would otherwise be losses and waste [13]. The topic of food loss and waste (FLW) often has been addressed, not so much as a question of incentives for resource use, but as a technical or physical measurement problem, with many studies measuring losses at different points along the food supply chain [15–20]. The more recent FLW literature has recognized alternative uses and markets for agricultural production [21–24], but as yet downplays the importance of optimizing individual decision-makers along the supply chain for determining the final allocation of product to various end points, including loss and waste. From an economic point of view, losses may be interpreted as simple outcomes of intelligent resource allocations, where decision makers at different nodes of the supply chain observe the costs and benefits associated with loss abatement efforts [23–26]. To better understand likely future FLW scenarios, analysts will also have to account for the introduction of new incentive structures. Advances in the bioeconomy will offer alternatives to what would otherwise be considered losses, while they will also likely encourage the diversion of resources from food production.

The analysis contributes both to the methodological toolkit to analyze the implications for resource use of the introduction of bio-based products, and to linking the literature on food loss and waste with that of the economic behavior of decision makers in the emerging bioeconomy. This line of study responds to the call for multidisciplinary research to improve the understanding—especially with respect to the importance of incentives

and markets—of the possible implications of advances in the bioeconomy, a call which was reflected in the recent Global Bioeconomy Summit (see https://gbs2020.net/). The analytical focus here is on the incentives to redirect some part of production from traditional markets to alternative markets, recognizing that the distribution of product quality is subject to farmer control and there is some portion of production that is (optimally) left unsold in any market (i.e., a loss or waste). The introduction of new alternative markets induces changes in resource use, which in turn would change the distribution of product quality. The proportion of production with zero economic value could be reduced via "biowaste valorization" [27]; that is, the diversion of what otherwise would be termed "losses" into economically profitable inputs, such as in the production of energy and/or the conversion into non-traditional, bio-based products. One contribution of the analysis is to present a framework to understand farmers' decisions regarding the distribution of product quality within a bioeconomy context, deriving the implications for the likelihood of farmer participation in new bioeconomy activities, for the volume of product sales in traditional and bioeconomy markets, and for product quality and loss levels.

The paper first presents an analytical framework and a basic model of quality and loss decisions, including alternative decisions regarding entering alternative markets. The research design begins with a standard optimization model in which the producer responds to relative price incentives in the allocation of production between two markets, a traditional and an alternative which carries with it some significant fixed cost of entry. The added value of our approach is to introduce explicitly three aspects of the producer's decision problem particularly important for anticipating outcomes in future alternative markets for biomass destined for transformation into a bio-based product. The first is that the quality characteristics important for the traditional market are unimportant for the alternative market. The second is that the quality characteristics important for the traditional market are often of a spectrum, with an associated range of prices, while the decision to participate in the alternative market depends not on the quality but the "total quantity" of low-quality production available, perhaps in the form of what would otherwise be losses or production diverted from traditional sales. If the total volume of low-quality production is small enough, even if it represents a large proportion of a farmer's total, then the farmer would be less likely to participate in the bioeconomy alternative. This yields the empirically interesting prediction that, as bioeconomy alternatives become available, both the low-productivity and high-productivity producer will tend to be excluded from bio-based supply chains. The third important aspect of the decision problem is that producers have control of the distribution of production levels across the quality spectrum. This leads to the prediction that as future bio-based alternatives become commercially available one would expect to see not only a shift of low-quality production to alternative markets but also a likely decrease in both quantities and "quality" levels of production available to consumers in traditional markets.

To illustrate the fundamentals of the analytical modelling effort, we present evidence of the distribution of quality and prices in the case of Chilean cherries, a fruit rich in dietary polyphenols and other compounds [28,29], where a significant proportion of harvested fruit is lost due to failure to meet export quality standards of processors, especially due to fruit size, a controllable quality directly linked to prices. The significant proportion of cherries unsaleable in traditional markets invites the future introduction of alternative bioeconomy uses [30]. We then turn to a numerical example to illustrate the implications of changes in economic incentives due to the introduction of an alternative use for otherwise lost production, showing that the opportunity cost of sales to traditional markets increases. Not only would sales to those traditional markets consequently fall, and the distribution of product quality, as measured by traditional consumers, shift downward, but both the very low productivity producer and the very high productivity producer would tend to be excluded from participating in the biobased alternative markets. We end with a discussion of the policy implications of the analysis, noting the potential negative impacts

on food buyers and other final consumers of traditional products due to the introduction of bioeconomy alternative uses for agricultural production.

## 2. Materials and Methods

### 2.1. A Model of Farmer Decisions Regarding the Distribution of Product Quality

The analysis begins with a previous line of literature of agricultural producer decision-making regarding the use of various inputs (e.g., pesticides) for damage and quality control [31–33]. It builds on this literature by linking new bio-based markets to decisions over quality and losses. This approach to understanding the incentives that induce participation in the bioeconomy and that simultaneously alter a product's quality distribution is also a contribution to better anticipating the implications of bioeconomy research and polices for agricultural markets. The basic conceptual framework to analyze the quality distribution of agricultural products, including product losses, and its relation to the decision to participate in bioeconomy activities is related to the literature regarding food loss and waste along the supply chain. From an economic point of view, food loss is defined in terms of food production that is not be used for further productive purposes; that is, "food waste" is defined as "the difference between the amount of food produced and the sum of all food employed in any kind of productive use, whether it is food or nonfood" [24] (p. 1152). The framework here uses this definition by considering product loss at the famer level because of decisions regarding the distribution of product quality. Our analytical model of production and quality-distribution decisions interprets production losses as that proportion of total production with a zero (shadow) price (i.e., without an economic value). This interpretation also permits connecting the more recent literature on FLW with a previous line of research related to "optimal culling" [34,35]. A product is discarded or culled when it does not meet some standard defined by certain attributes, such as size, maturity, etc. [34]. The literature on optimal culling often emphasizes industry coordination, perhaps via "marketing orders" or some other government-enforced group regulation. The analyses of the implications of culling are usually set in the context of control over total market quantities supplied with the purpose of increasing industry profits through collective monopolistic behavior, or in the context of building buyers' long-term confidence where group reputation is a common property resource subject to free riding [34–36].

Figure 1 shows the allocation of agricultural production based on the farmer's decision regarding quality attributes. A farmer makes decisions with respect to inputs, effectively deciding the quantity and quality of his production. The farmer selects the level of inputs, such as fertilizer, water, land, and energy to produce a certain quantity of production. The total production, represented by $(y)$, contains various subsets with a range of attributes, which lead to a sorting of production by expected prices into different quality levels. The ex-ante optimal range or distribution of quality would determine "expected" net revenues. Many decisions about input use related to the final distribution of quality would be taken prior to imperfectly predictable events, such as weather and pests, and so these decisions would likely be influenced to some degree both by the timing of possible external factors and by the risk attitudes of farmers to random quality and damage control outcomes [33]. In Figure 1, realized quality levels are ranked from 1 to M, where 1 represent the highest expected price, or the highest quality, and M the lowest quality that can be sold for a positive price. In the figure, the farmer sends his production to different markets, perhaps for different uses. For example, in the case of fruit, some production is sold to high-end restaurants and niche boutiques and corner shops for higher-income consumers, other production is aimed at big-box chain stores for lower-income mass sales, and some portion of production goes to juice processors. In the figure the parameter $\theta_i$ represents the proportion of the total production, with a quality level $i$, which is sold at $P_i$ in the market $i$. More specifically, $\theta_1$ is the proportion of production with the highest quality, sold at the highest price $P_1$, and the sum, $\sum_i^M \theta_i$, is sold to M traditional markets. The figure also shows another proportion of total production, $\theta_{M+1}$, that does not meet the requirements for any traditional market with a positive price, on account of its low quality.

With the introduction of an alternative bioeconomy use for the farmer's production, there is a possibility to devote some if not all of $\theta_{M+1}$ to this new use, represented by $\theta_b$. This proportion can be sold to other users, in which case a price, $P_b$, will be observed, or used by the farmer himself, in which case $P_b$ is a shadow price. In sum, $P_b$ can be considered the value of the marginal product of the non-traditional proportion not accounted as a loss.

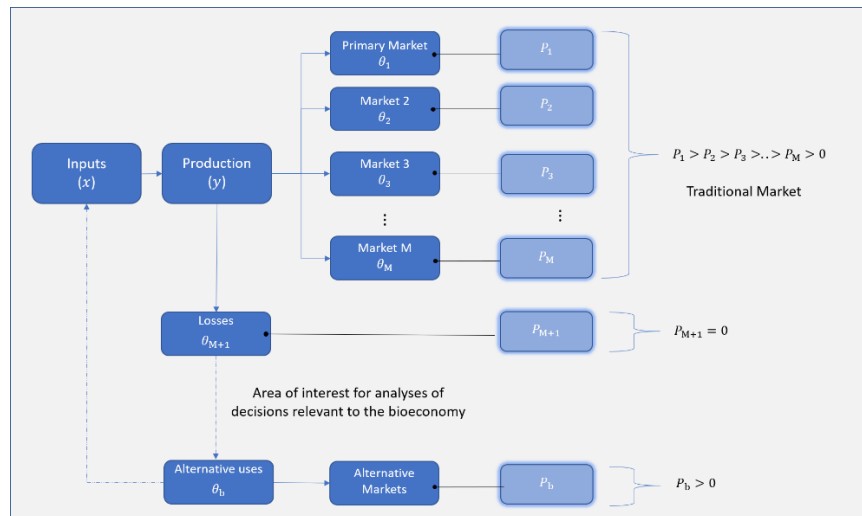

**Figure 1.** Analytical framework for farmer product quality decisions. A farmer's decisions over input use determines both the level of expected total quantity produced ($y$) and expected proportions of various quality levels ($\theta_i$) associated with M traditional markets, ranked according to expected prices, $P_1 > P_2 > \ldots > P_M$. Without alternative uses, the proportion of "losses" ($\theta_{M+1}$) has a price of zero. However, with an alternative, bioeconomy outlet, a proportion ($\theta_b$) can be diverted other (bioeconomy) uses with a positive (shadow) price $P_b > 0$.

In this framework, the proportion of total production that does not meet quality requirements could be diverted to new usages in the bioeconomy. Production decisions at the farmer level will lead to some optimal level of low-quality product and therefore some optimal level of losses. Optimal loss is defined here in the standard economic sense that the marginal cost of reducing loss further would exceed expected marginal benefits. The approach here is to evaluate decisions concerning losses in terms of farmer control over product characteristics that translate into expected prices. Quality is to be understood in this context as some set of characteristics that maps onto a single dimension, price. Prices that vary by quality could be determined via various mechanisms. For example, relative quality-varying prices could be determined in negotiations between buyer and seller, or according to some price schedule defined by buyers or some market intermediary. Relative prices of differing qualities could be the result of market forces in distinct destinations for final product sales (e.g., export quality versus domestic quality, á la Alchian—Allen effect [37,38]).

We take the approach that input decisions influencing the distribution of quality will be evaluated based on how quality is reflected in relative prices. In short, better quality means higher prices. In agricultural production quality can be represented by a discrete or a continuous variable. As a discrete variable, the use and non-use of a specific method in product production, for example, can define the quality attribute (e.g., caged vs. free-range chickens), as is discussed in [39]. On the other hand, when a method or process affects a distribution of various qualities, one can model this distribution as a continuous variable. For example, a type of variety of fruits is considered as discrete quality attribute while continuous attributes include color, size, texture, flavor, nutrients, etc. [31]. In this latter case, the farmer, or some intermediary, can sort production for sales in a variety of markets, with the result that some portion of production will not be sold (i.e., lost).

Our model here begins with a crop with a distribution of levels of characteristics or attribute, $a$, which determine a real-value, continuous index, or grade level, $z = z(a)$, which corresponds to the ranking of prices from highest to lowest. We abstract from possible timing of input use decisions and from external shocks (such as weather and pests) during the growing season and assume that each farmer makes decisions to control the quality distribution via the use of inputs to control the attributes (following the modeling frameworks in [31,32]). Without loss of generality, the continuous quality level, z, can be confined to the interval 0 to 1. Let $f(z)$ represent the distribution function of the quality variable given the decision over the input use, where $\int_0^1 f(z|\beta)dz = 1$, where $\beta$ represents a set of parameters (moments) that define the distribution and are controllable via input decisions by the producer. Note, that this distribution refers not to a probability distribution but to a distribution of continuous quality outcomes.

In the case of a finite number of discrete prices for products falling within distinct groupings of ranges of the quality scale, one can model the sales of each quality group in terms of the proportion of sales to each market $i$. Let this proportion be represented by $\theta_i$, where the index $i$ follows the rank of the markets in terms of price ($p_i > p_{i+1}$), and the sum of the percentage of production assigned for $M$ markets plus the percentage unsold (losses) sum to one. Let $\theta_{M+1}$ represent the proportion unsold, and so $\sum_{i=1}^{M} \theta_i + \theta_{M+1} = \theta_s + \theta_{M+1} = 1$, where $\theta_s$ represents the proportion of total production sold. The proportion of sales to each market is defined by a quality range defined within the interval of critical quality bounds $[c_{i+1},\ c_i]$:

$$\theta_i = \int_{c_{i+1}}^{c_i} f(z|\beta)dz \tag{1}$$

The proportion of food unsold, $\theta_{M+1}$, is that which does not meet the minimum quality attributes for any market:

$$\theta_{M+1} = \int_0^{c_M} f(z|\beta)dz \tag{2}$$

Now consider the total agricultural production, represented by $y$, summed overall quality levels, and a vector of inputs, $x$, which affect the total level of production, the product's quality attributes, and therefore the proportions of sales for each market. In implicit form, farmer's production function is given by the restriction $T(y, x, z, \theta) = 0$, where $\theta$ is the vector of $M + 1$ proportions. From this technology constraint, and a vector of input prices ($w \gg 0$), the farmer minimizes the costs of producing $y$ units with a given distribution of qualities:

$$C(\theta, y, w) = \min_x \{ w'x \mid T(y,\ x,\ z, \theta) = 0 \} \tag{3}$$

The cost function is determined by the profile of percentages of production assigned to different markets. When deciding the proportion of the product going to each market, the farmer observes the costs associated with quality requirements. With a vector of agricultural product prices, $p \gg 0$, there is a corresponding price for each quality attribute, where the average product price received across sales is $\overline{P} = \sum p_i \theta_i$. For simplicity at this stage of model development, we consider the static situation, abstracting from uncertainty in agricultural production and prices. As in the case of production analysis more generally, for future empirical work one would have to take into account the influence of decision timing, external conditions (especially weather) on quality and quantity of production, the possible non-linearity of impacts on profits, and farm risk aversion [33]. However, as a first step, we distill the analysis to the maximization of net benefits during a single period. Considering prices as given, the farmer's problem would be to maximize profit ($\pi$) by choosing the mean, precision, and the production level:

$$\pi(p, w) = \max_{\theta, y} \{ \overline{P}y - C(\theta, y, w) \}, \tag{4}$$

leading to decision rules for both optimal total production, $y$, and the optimal proportions of the quality categories. This optimization framework implies that there is an optimal level of production at each quality level and an optimal level of losses. The farmer would choose total production such that the additional cost of increasing production would not exceed the average price received from sales across all quality levels.

### 2.2. The Case of Three Qualities: High, Low, and Loss

One could delve into the technical microeconomic analytical details of the farmer's decision model, but the basic implications of the introduction of a bioeconomy alternative for the quality proportions, $\theta$, and what would otherwise be product losses can be illustrated via a simplified model. Consider the proportion of production of three quality levels ($\theta_1$, $\theta_2$, $\theta_3$) with three potential markets and prices ($p_1$, $p_2$, $p_3$): high quality, low quality, and an alternative market. How do farmer decisions change with the introduction of this new outlet for product sales, offering a positive but lower price compared to traditional? How does scale or productivity enter into the decision to participate in the bioeconomy activity, given that a farm's size and productivity level are relevant to technology adoption in agricultural [40,41]. Decisions concerning product quality are affected if the farmer chooses to participate in the new market. The farmer evaluates if it would be more profitable to sell what would otherwise be losses in the new market. In the absence of any cost to accessing this alternative market, clearly the farmer would choose to participate in the bioeconomy, implying an alteration in input use and in the level of total production and sales to traditional markets.

However, there are likely costs associated with participating in the bioeconomy, such as investment in on-farm collection, storage and quality maintenance, transportation, etc. The most generic model of a farmer's decision to adopt or not an alternative business strategy is based on straightforward utility maximization. Let $d$ be a binary indicator, where $d = 1$ represents the case where the net benefits derived from adopting a production plan that takes advantage of an alternative market is greater than the case ($d = 0$) where the farmer produces only for traditional markets. The decision maker's utility $U_d(A, Y)$ in some period (say the crop cycle, perhaps a year) depends on the farm's and the farmer's attributes, $A$, and the farmer's optimal production and quality decisions, represented by the vector $Y_d$, which in turn depend on the optimal strategy selected. The farmer participates in the bioeconomy market if: $U(A, Y_1) > U(A, Y_0)$.

The farmer takes output prices as given, where prices correspond to given intervals of critical quality bounds, $[c_{i+1}, c_i]$, and again $p_1 \geq p_2 \geq p_3$. The adoption decision becomes interesting with the introduction of some additional cost $f$ to participate in the alternative market (and fixed with respect to other production decisions). In this single-period model, the fixed cost, $f$, might be associated both with the amortized expensing of the upfront investment costs of bioeconomy-specific business assets over the life of those assets, and with any other costs linked to participating in the alternative market, but which are unrelated to the volume of production and quality levels (e.g., maintenance of storage facilities). For simplicity, we set aside considerations related to uncertainty and risk aversion. The farmer maximizes profit by choosing to enter into the alternative market, $d = 0$ or $d = 1$, and by adjusting the proportions of production destined to the high-quality and low-quality markets, $\theta_1$ and $\theta_2$. The remaining proportion, $\theta_3$, is loss, if $d = 0$, or sold in the alternative market, if $d = 1$. The simplest representation of the farmer's utility maximization problem reduces to maximizing net income in the period:

$$\pi_d^* = \max_{\theta_1, \theta_2, y, d} \{y(p_1\theta_1 + p_2\theta_2 + dp_3(1 - \theta_1 - \theta_2)) - C(\theta_1, \theta_2, w, y) - fd\} \tag{5}$$

This problem can be rewritten to highlight that the introduction of the alternative market reduces the incentives for sales into traditional markets:

$$\pi_d^* = \max_{\theta_1, \theta_2, y, d} \{y[(p_1 - dp_3)\theta_1 + (p_2 - dp_3)\theta_2 + dp_3] - C(\theta_1, \theta_2, w, y) - fd\} \tag{6}$$

The farmer enters the bioeconomy if the benefits of doing so exceed the opportunity cost: $\pi_1^* \geq \pi_0^*$. Note that, if so, $(p_i - dp_3) = (p_i - p_3) < p_i$, for both traditional market quality categories, i = 1, 2; which is to say that the incentives to produce for those markets are, at the margin, lower.

*2.3. A Graphical Illustration*

Figure 2 summarizes intuitively the model presented above. In the case of sales exclusively to traditional markets, the optimal distribution of the continuous quality index is given by curve A. A specific quality value maps onto a corresponding net effective price that the producer would receive by selling a unit of product at that quality level. This net price could be zero, and the farmer would not sell into the traditional market. This lowest quality/price, P1, represents the lowest quality that could be sold, and the proportion of production with lower quality levels would be losses.

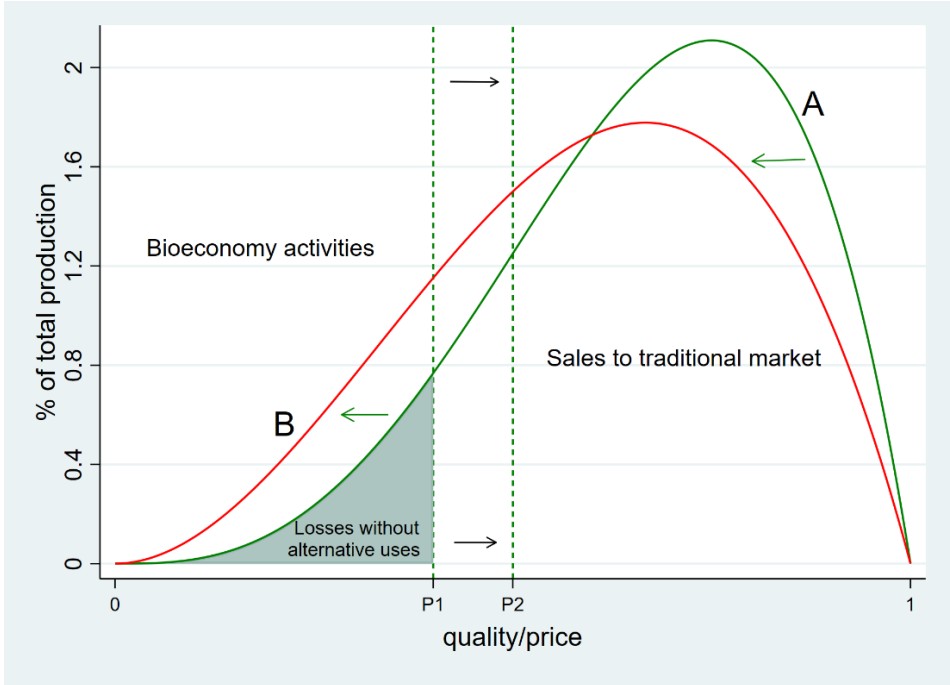

**Figure 2.** Optimal distribution of agricultural product quality conditional on output and input prices. Losses represent the percentage of total production which does not meet traditional market requirements. P1 is the critical quality/price point where traditional market products become sellable. Curve A represents the quality distribution of an agricultural product when losses are not destined for alternative uses. Curve B represents the distribution of quality when bioeconomy activities are available. There are two main effects of the introduction of an alternative market for what would have otherwise been losses. The increase in the critical price level from P1 to P2 (indicated by →) induces less quantity sold to traditional markets even with the original quality distribution. The second effect is a shift leftward in the quality distribution (indicated by ←), further reducing sales into the traditional market.

Figure 2 also shows the introduction of the bioeconomy alternative, where losses can be reduced via waste valorization. The alternative market offers a positive price, independent of quality level, corresponding to the quality/price level P2 in traditional markets and higher than P1. This higher, new price makes the sale of what would have otherwise been losses an economically attractive alternative. There will be, therefore, due to a revalorization of what were otherwise losses, an increase in the opportunity cost of raising agricultural product quality. One can decompose the impact on sales and quality decisions into two effects, both in the same direction, as shown in Figure 2. First, the

increase in the critical price that would induce sales to the traditional markets raises the opportunity cost of such sales for lower quality levels. That is, sales to traditional markets would fall even if farmers maintained the distribution of quality constant. Second, the farmer's decisions would opportunistically respond to altered incentives, adjusting input levels and the product quality distribution to take advantage of the new, alternative market. The marginal returns to elevating quality and to abating losses would fall, and so the distribution of quality should shift leftward, as in curve B, the exact change in the quality profile depending on technology and prices. In any event, sales into traditional markets would decline; and if enough farmers adopt alternative bioeconomy activities one would expect overall price increases and overall quality decreases in these traditional product markets. In the case of staple foods, this might have worrisome consequences for the mass of consumers, particularly for those with modest incomes. The final (and at present indeterminate) equilibrium results in domestic and international food markets due to the widespread adoption of bioeconomy activities are well beyond the scope of the present paper, but such an analysis would be well worth developing in future research especially in the light of possible unintended negative consequences for low-income consumers.

## 3. Results and Discussion

The conceptual framework and algebraic model above can be made more concrete with a real-world illustration of the distribution of quality and resulting loss levels, in this case the export-oriented production of Chilean cherries. Recently bioeconomy researchers have shown interest in recovering polyphenols from cherries that have would otherwise have little or zero market value, especially due to their size. In particular, efforts are underway to explore methods of processing antioxidant polyphenol-rich extracts from low caliber fruit [30]. This is a particularly relevant line of research for potential alternative uses for cherries, especially in the case of Chile, where cherry production has grown rapidly and significantly over the course of the last decade. In fact, currently cherries represent the most valuable fruit exported from Chile, exceeding table grapes and reaching over $1.4 billion US dollars in export value [42].

We make use of quantity and quality data at the processing plant reception level from a Chilean cherry processing-exporting company which acquires its fruit from many growers in its local district. The data are for 50 growers located in the main cherry production zone near the town of Curicó in Chile. Cherry exporters, marketing to demanding standards in China, Europe, and the United States, make use of sophisticated and highly mechanized inspection and classification processes to sort and package cherries according to quality attributes such as caliber, color, firmness, soluble solids, and fruit defects. The information we use here reflects the 2018–2019 crop year. We focus on fruit caliber to illustrate the distribution of quality and the proportion of losses (at the processor level), which represents a first-order approximation of the proportion of production potentially available to some bioeconomy alternative use, such as in antioxidant extraction.

We then turn to a simplified numerical example of farmer decision model to demonstrate the relationship between adopting an alternative bioeconomy activity, product-quality decisions, farm productivity, and loss levels.

### 3.1. The Distribution of Quality Levels and Prices in Export-Oriented Chilean Cherries

Figure 3 shows the proportion of cherries during the season by their classification according to the size of the fruit. For example, the third bar from the left represents the so-called "large" category and shows the percentage of all cherries received by the processor-exporter that are between 22 and 24 mm in diameter. The "local sales" category represents unexported cherries destined for local buyers, and usually of a smaller size than "large." These categories are the quality dimension that has a direct relationship to prices received by producers. Figure 3 also shows the median and average prices received during the period 2018–2019 by quality class. The maximum and minimum prices received are also shown. Growers can influence the distribution of fruit sizes and other attributes

through nutrient use, pruning, thinning, and other practices, but the harvest contains a wide range of cherries of different attributes and are sorted according to the exporter's assessment of buyers' willingness to pay. The size classifications shown in Figure 3 are according to marketing names which identify the final product for wholesalers, retailers, and consumers. These size classifications map directly onto the prices farmers receive, the larger the fruit the higher the price. Note, however, that median prices increase as quality increases, but the range between minimum and maximum prices is large due to the timing of sales during the season and other attributes, such as the ripeness of the fruit. Very early and very late in the season, cherries command significantly higher prices, and during the height of the harvest, prices are lower. Fruit that is "too ripe" to endure longer-distance transport in perfect condition also receive lower prices. In concrete terms these minimum prices of each cherry quality for the 2018–2019 season are the minimum prices for which it would have been advantageous to begin to sell biomass to buyers seeking the raw material for antioxidants. For example, if the antioxidant market were to have offered a price above US$0.10, growers at some point in the season would have not only been willing to deliver the biomass in the loss category in Figure 3 to the alternative market but would have also been willing to switch deliveries of low quality but saleable cherries destined for traditional local buyers. An antioxidant market price of above US$0.70 would have led to a delivery to the bio-based alternative of all of the biomass in the "loss" and "local buyer" categories. An antioxidant market price of above US$1.85 growers at some point in the season would change even deliveries of exportable "Large" cherries destined to traditional foreign buyers.

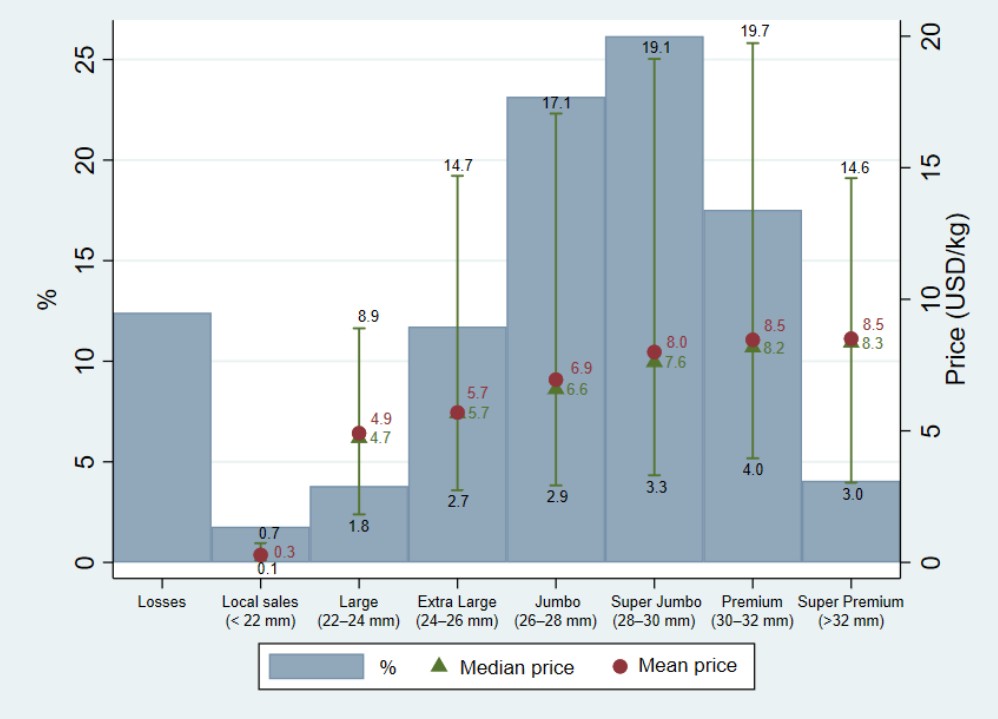

**Figure 3.** Cherry size distribution and associated prices for each size classification. The graph displays the average, median, maximum, and minimum values of cherry prices according the size category. The "local sales" category represents unexported cherries for local buyers. Red circles represent mean prices, green triangles refer to the median price and the numbers at the top and the bottom of the red lines represent the maximum and minimum price paid for each size category.

The costs related to processing, packaging, refrigeration, transport, and other handling activities are independent of cherry quality and final price paid; and, given buyers' preferences and willingness to pay according to size, cherries of sizes significantly below

22 mm are considered unprofitable (the Alchian–Allen effect). The categories "losses" and "local sales" therefore show the accumulated sum across all sizes significantly less than 22 mm. Any cherry, otherwise undamaged and classified as "large" or bigger are sold; that is, 22 mm approximates the trigger level (P1 in Figure 2) between the traditional market and losses (although there are some sales to local buyers). As can be appreciated from Figure 3, unexportable fruit amounts to approximately 14.2 percent of delivered cherries, a significant proportion which, one should note, does not include those quantities lost in the orchard, at or prior to harvest. A small proportion of this fruit is economically useful to other local buyers. Total unsaleable—complete losses—remain at 12.2 percent of delivered cherries. This significant proportion, which is likely similar to other exporters, is a first-order approximation of the biomass available of cherries for bioeconomy products. The introduction of, say, a market linked to antioxidant polyphenol extraction would begin with this proportion of current losses as a base.

Is the distribution of cherry quality significantly influenced by farmer practices? The export cherry data available are not yet accompanied by details of grower production decisions, but the processor-exporter sorts farmers, based on their assessment of the producers' "skills" and production strategies, as reflected in the ability to consistently of delivery larger volumes of higher-quality fruit in previous seasons. Growers are ranked into three skill reputation categories, A, B, and C, from most reliable to least. Figure 4 shows the percentages of quality levels by these types of producers for the 2018–2019 season. Loss proportions clearly decline according to skill level, producers of the C-type having the greatest loss levels and the A-type the smallest, and the proportions of the highest quality level clearly increase according to skill. During the 2018–2019 season, the distribution of the quality measure, and therefore also of prices received, shifts rightward according to the skill classification based on previous seasons. While further analysis at the level of input use and orchard management would be required, the evidence here and from recent studies (e.g., [17,43]) strongly suggests that the quality distribution leading to losses, is controllable by producers. One would suspect therefore that the introduction of other uses for non-exported cherries, such as the extraction of antioxidant polyphenols, would reduce total export volumes and average export product quality.

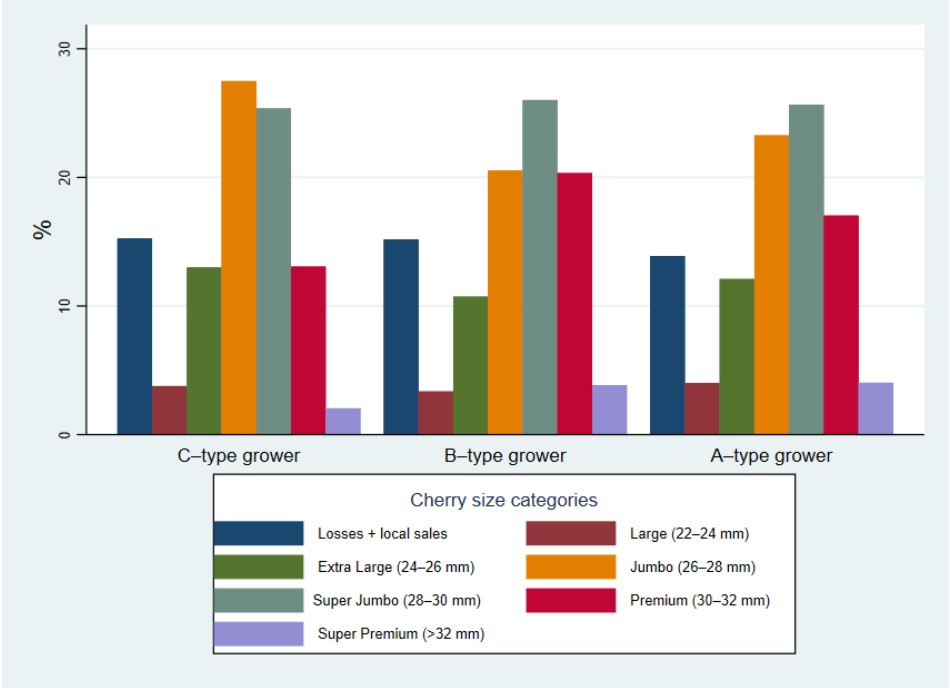

**Figure 4.** Distribution of cherry size by type of grower. Cherry size categories are ranked from Losses to Super Premium.

*3.2. A Numerical Example*

One empirically interesting prediction deriving from the general analytical approach to modeling producer control of the distribution of production levels across a range or spectrum is that as future bio-based alternatives become commercially available one would expect to see not only a shift of low-quality production to alternative markets but also a likely decrease in both quantities and quality levels of production available to consumers in traditional markets. Another implication that is implicit but not immediately obvious from the basic algebraic representation of the farmer's decision problem presented above is the relationship between farmer productivity and the decision to participate in the bioeconomy activity. Because quality levels important for the traditional market range across a spectrum, each quality level carrying an associated price, one can show, in general theoretical terms, that the decision to participate in the alternative market would be dependent on the total quantity of low-quality production available. Such low-quality production would be both in the form of what would otherwise be losses or in the form of a diversion of lower-quality quantities from traditional sales. If total production of low-quality product is a sufficiently small quantity, even if it represents a large proportion of a farmer's total, then the producer would be less apt to participate in the bioeconomy alternative. This yields an empirically important prediction: as bioeconomy alternatives activities become available to farmers, both the low-productivity and high-productivity producer will tend to be excluded from bio-based supply chains. Mid-productivity producers would tend to have a representation in bioeconomy activities proportionally greater than their share of the total farmer population.

We turn in this section to a numeric example to illustrate more clearly these implications for farmer production quality decisions of introducing an alternative bioeconomy market. In this toy model, we posit that the farmer faces two options as described above in Section 2, not participating ($d = 0$) in the bioeconomy activities and leaving some portion of production unsold as a loss, or participating ($d = 1$), and being able to sell what would otherwise be a loss, but incurring some fixed cost, $f$, associated with the new activity. We simplify the model by considering a single traditional market, where the controllable proportion $\theta$ represents the percentage of the product that complies with the quality standards of all traditional markets, sold at the average price $P_t$, and where the proportion $(1 - \theta)$ represents the proportion of low-quality product, and is either the unsold product (losses) or the product sold in the alternative market. The high-quality proportion $\theta$ will depend on optimal input decisions and be conditional on the opportunity cost of sales to the traditional market, which in turn depends on whether or not the bioeconomy alternative market is profitable. We abstract from the complication of disentangling the joint decisions regarding both production level, $y$, and high/low quality distribution by fixing total productivity. We can then perform a comparative-statics exercise showing the relationship between productivity level, the decision to participate and the decision over the quality proportions.

If the farmer decides not to participate in the bioeconomy activity, the price of the proportion unsold is of course zero; if participating, the farmer pays a fixed cost of $f$ and the proportion of the product not meeting the standards of the traditional market can be sold for a positive price ($P_b > 0$). Let $C(\theta)$ represent a cost function relating optimal input and management expenses associated with farmer practices that attain the proportion, $\theta$, of the high-quality production saleable in traditional markets. Production costs are increasing in $\theta$, and if the optimal level of this high-quality proportion is positive, but less than one, costs are increasing at an increasing rate. In this numerical example, we take costs of high-quality production to be a simple quadratic $C(\theta) = \frac{k}{2}\theta^2$. We assume the farmer is a price-taker and set $P_t = 3$, $P_b = 2$ and $k = 6$.

The algebraic details of the farmer decision rules in this numerical example are given in Appendix A, but are shown graphically below. In the first scenario, where the farmer decides not to participate in the alternative market ($d = 0$), the optimal proportion of production sold in the traditional market is proportional to the traditional price, increasing

in productivity, $y$, and the optimal level of losses is simply the remainder, $(1 - \theta^*_{d=1})$. In the second scenario, where the farmer decides to participate ($d = 1$), the marginal opportunity cost of traditional sales increases; the optimal level of production sold in the traditional market is now proportional to the net gains of traditional market sales over what can otherwise be obtained in the alternative market:

$$\theta^*_{d=1} = \frac{(P_t - P_b)y}{k} < \theta^*_{d=0} = \frac{P_t y}{k} \tag{7}$$

The optimal proportion of production sold in the alternative market, receiving price $P_b$, is simply the remainder, $(1 - \theta^*_{d=1})$. The farmer chooses to participate based on comparing relative net revenues, including the fixed costs, associated with the two scenarios, and relative profits will depend on the farmer's productivity level, $y$. From a point of low productivity, an increase in the parameter $y$ increases the marginal benefits of producing additional units of the high-quality product saleable to the traditional market, and thus also a decrease in losses. The relationship between profits and productivity is not merely positive but convex, because the farmer can take advantage of an increase in $y$—or mitigate a decrease in $y$—by appropriately adjusting the optimal quality proportion $\theta^*_d$ in both scenarios. The marginal benefits, however, of adjusting the quality proportion saleable in the traditional market are less when the alternative market is available, and the convexity of net revenues with respect to productivity will be less pronounced with the participation in the bioeconomy activity. The degree of convexity will depend on $(P_t - P_b)$. This might appear at first an economist's technical modelling detail, but it reflects the important fact that the benefits of assuring high-quality production "fall" with the bioeconomy alternative. Net benefits would be less sensitive to an increase in productivity because the opportunity to sell into the alternative market effectively "increases" the marginal cost of accessing the traditional market. The cost of this access does not merely include the cost of the inputs required to attain the necessary quality standards for an additional unit sold into the traditional market, but also the forgone revenue of "not" selling that unit into the alternative market. This reduction in the sensitivity of profits to productivity, when the farmer does participate in the bioeconomy activity, also leads to the conclusion that, in the presence of fixed costs to participation, both the very low-productivity farmer and the very high-productivity farmer would be less likely to make use of the alternative market. In both cases, the optimal levels of the low-quality production are insufficient to pay for the fixed costs of the new activity.

　　Figure 5 explores the introduction of alternative markets considering both a range of fixed costs and productivity levels. The figure shows the relationship between profits and productivity levels without participation in the bioeconomy activity and with participation under three possible levels of fixed costs ($f$ = 1.00, 1.25, and 1.45). After some algebraic manipulation one can find the critical levels of productivity, conditioned on fixed costs, where the farmer would be indifferent to participate in the alternative market or to participate exclusively in the traditional market but also accept product losses (see Equation (A2)). With zero fixed costs, the farmer would always participate in the alternative activity, because the farmer could always, at minimum, exclusively access the traditional market and accept some losses; therefore, having the alternative market as an option would only have a potential upside. However, as fixed costs grow, the decision to participate in the bioeconomy involves a comparison of relative profits. For three given values of the fixed cost, $f$, Figure 5 shows the critical levels of productivity dividing participation from non-participation. These critical points are where the profit-productivity curve under non-participation intersects the profit-productivity curve under participation. Note that the participation zone between these critical points shrinks with higher fixed costs. The low-productivity farmer, while producing a higher proportion of low-quality product, has an insufficient total quantity of potential deliveries to the alternative market to recover fixed costs. The high-productivity farmer, while producing a large quantity of potential deliveries to the alternative market, finds it profitable to invest in attaining the

traditional market quality standards. Therefore the high-productivity farmer would optimally produce an insufficient level of losses that could potentially recover the fixed costs of entering the alternative market. Only the mid-range productivity farmer would optimally have a sufficient quantity of low-quality product to sell to the alternative market that could recover the fixed costs of adopting the bioeconomy activity. As Figure 5 illustrates, given a fixed cost $f = 1.45$, the farmer would be indifferent to entering the bioeconomy activity at productivity levels of 1.22 and 1.77. The farmer would optimally decide to participate in the alternative market when productivity falls within those critical, indifference point $(1.22 < y < 1.77)$.

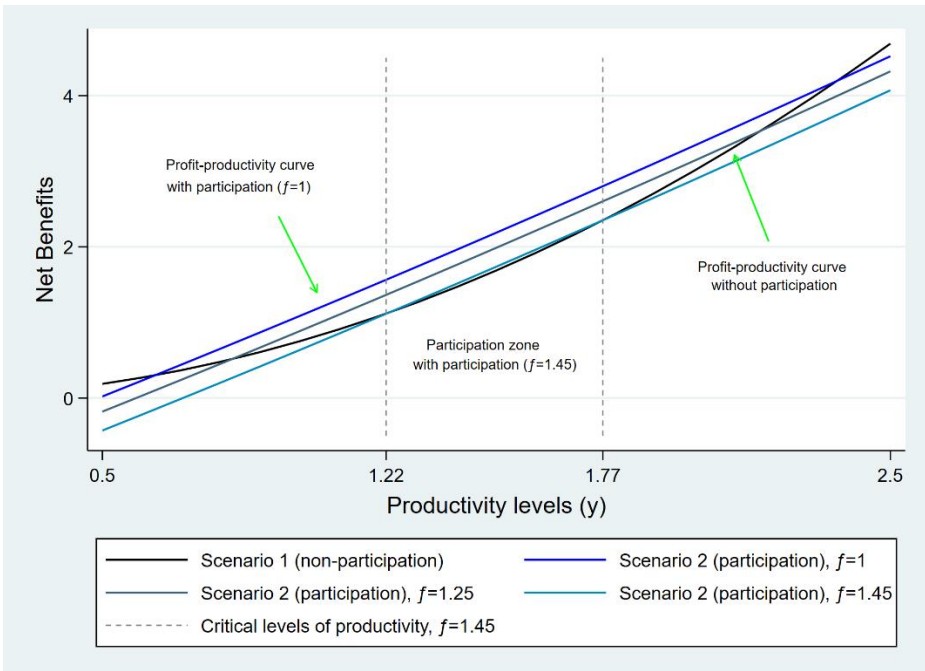

**Figure 5.** Productivity levels and the farmer's decisions to participate in the bioeconomy activity. Three values for the fixed cost of participation (scenario 2) are $f = 1.00$, 1.25 and 1.45.

To understand more clearly the incentives driving the decision to participate in bioeconomy activities, Figure 6 shows the relationship between sales to the traditional market and potential product losses, given the fixed costs $f = 1.45$ of participation. Note that increases in the level of a farmer's productivity, in the absence of the alternative market, leads to a higher optimal proportion of the product meeting the standards for the traditional market, and leads therefore to lower optimal losses. As noted above, greater productivity lowers the effective marginal cost of meeting quality requirements and the total quantity of production and the proportion sold to traditional market is greater. Comparing across productivity levels, as total productivity becomes sufficiently high that the associated level of optimal losses makes paying the fixed costs of the bioeconomy activity profitable, the farmer would optimally make a discrete shift to a "lower" quality regime where the marginal revenue of one more unit sold to the traditional market is now not $P_t$ but $(P_t - P_b)$. Sales to the traditional market fall as the farmer moves from an exclusive reliance on the traditional market and finds that the marginal benefit of meeting that market's quality standards is now lower.

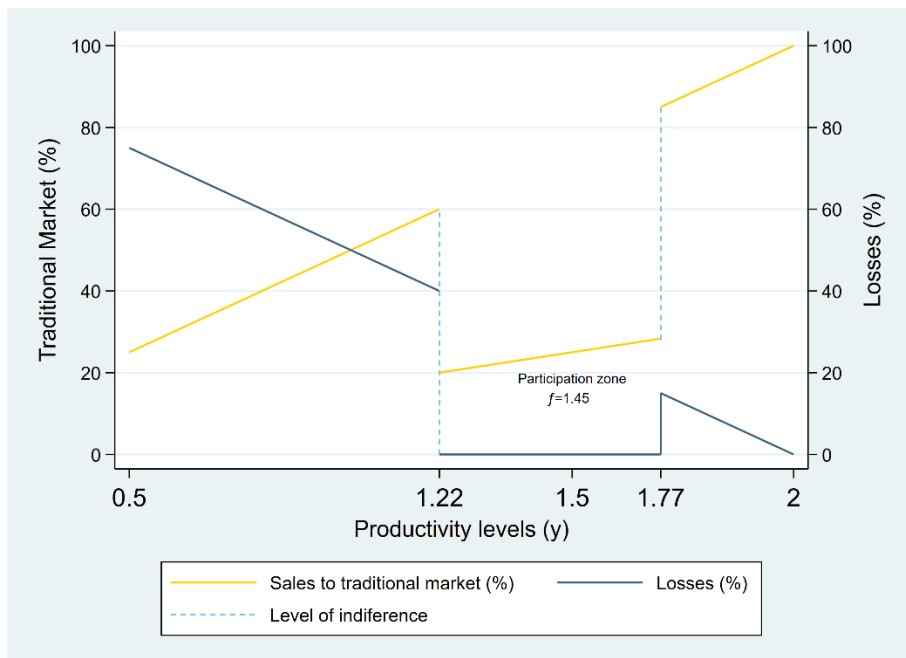

**Figure 6.** The proportion of total production destined to the traditional markets and the loss proportion vary as the level of productivity increases. The upward-sloping (yellow) line represents the product quantity (%) serving traditional markets, and the downward-sloping (solid blue) line refers to loss proportions. Given fixed costs $f = 1.45$ of adopting the bioeconomy activity, producers with productivity levels, $y$, between 1.22 and 1.77 find attractive the alternative market.

The aggregate impacts on equilibrium market quantities and prices in traditional markets with the introduction of bioeconomy-related alternative activities would depend in major part on a combination of factors related to fixed costs, the distribution of productivity across producers, and the distribution of the sourcing of total supply across producers of various productivity levels. If fixed costs of engaging in the bioeconomy activity are high, and the preponderance of total supply derives from many, low-productivity (small-scale) farmers, or from a few high-productivity (large-scale) farmers, then the impact on the traditional market for, say, food would be relatively modest. However, if fixed costs are low and the preponderance of total supply derives from mid-productivity (mid-scale) farmers, then the impact on the traditional market could be significant and, in the case of staple foods, of some concern to policy makers.

### 3.3. Practical Implications and Consequences for Public Policy

There are some practical implications and consequences for public policy of the model and numerical example that are worth noting here. Sustained success in bringing new bio-based alternatives to market would depend on the scale of the biomass resource available from primary producers and consequently on the unit costs of accessing that biomass. One implication of the foregoing analysis is that one would expect to see investors in the bioeconomy targeting bio-based products that would make use of an input supply chain deriving from many mid-productivity farmers. This would tend to exclude smaller farmers and regions with less sophisticated farming systems from the focus of bioeconomy R&D and market-development efforts. Furthermore, one would expect that the bulk of farmers in less-developed countries would be unable to access, at least in the near term, the direct economic benefits of private and public investments in developing bio-based products. There might be, however, indirect benefits to lower-productivity producers in less-developed regions arising from higher-productivity competitors in more-developed regions shifting their supplies of traditional, tradeable goods across the quality spectrum towards bioeconomy alternatives.

A related implication is that, if there is a public policy interest in broadening the scope of beneficiaries of taxpayer supported activities to support bioeconomy development, some consideration might be given to assisting lower-productivity farmers in either covering the costs of initiating participation in alternative markets or spreading the costs over a larger volume of potential biomass for those markets. One way of encouraging participation in the bioeconomy alternative is to simply subsidize any fix investments associated with entry into the new market. Because, as shown in the analysis above, the ability to gain from participating in the alternative market depends on the "total" quantity of low-quality production available (from the traditional buyer's perspective), even if it represents a large proportion of an individual farmer's total, another way of encouraging participation would be to develop systems where farmers can coordinate their potential supplies of biomass for alternative uses. Such coordination might be achieved via cooperatives or market intermediaries, with the assistance of governments and bioeconomy specialists to reduce the transactions costs usually associated with organizing many smaller-scale operations. Policy makers might consider reorienting some aspects of the research efforts into the logistical problems associated with channeling biomass from primary production to final transformation into bio-based goods. Research efforts might be better aimed at the coordination of smaller-scale, lower-productivity farmers. Market intermediaries (as in the Chilean cherry example discussed above), who are already in the business of coordinating many producers for traditional supply chains, might be a nexus by which bioeconomy alternative outlets could be made available to the lower-productivity farmer.

The role of scientists and engineers in developing new bioproducts and making efficient the production and processing of the raw materials for these new products is effectively both to reduce the fixed costs of participating in bioeconomy activities and to raise the marginal benefits of sales into the associated alternative markets. As advances in the life sciences lead to more widespread profitability in market activities involving non-traditional goods and services, the more consequential will be the bioeconomy for the use of resources at the level of primary producer, and the greater will be the impact on traditional markets.

## 4. Conclusions

Several studies have highlighted the likely promise of advancing bioeconomy strategies, not only to boost a variety of economic activities, but to reorient the way in which goods and services are produced and reliant on nonrenewable and renewable natural resources. While the bioeconomy approach shows high potential for some farmers and other economic actors, it is nevertheless important to consider the market-mediated implications of individual decisions regarding new alternative activities in order to better anticipate the consequences of how the bioeconomy will unfold and to formulate related policies. As Zilberman et al. notes, "One of the challenges of expanding the range of products produced by agriculture is increasing the productivity of agricultural production to make food affordable globally while also expanding the range of products produced by farms" [5] (p. 101).

We have focused here on the links between a farmer's decision regarding the distribution of quality (including loss levels) and the decision to participate in new bioeconomy activities. The added value of our approach, based on a standard model of farmer optimizing behavior, is to introduce explicitly three aspects of the producer's decision problem important for anticipating the performance of future alternative markets for biomass: relevant quality characteristics differ between traditional and alternative markets, quality levels important for traditional markets are associated with a range of prices, and producers have control over the distribution of production levels across the quality spectrum. One implication of this approach is that the decision to participate in a bioeconomy alternative market will depend on the total quantity of low-quality production available. Therefore, one would expect to see that, as bioeconomy alternatives become accessible in the future, both the low-productivity and high-productivity producer will tend to be excluded from bio-based supply chains. Furthermore, as future bio-based alternatives become commer-

cially viable one likely will observe a shift of low-quality production to alternative markets and a decrease in both quantities and quality levels of production available to consumers in traditional markets.

The results underline that the introduction of new alternative markets induces changes in resource use, which in turn change the distribution of product quality, including the proportion of production not meeting traditional quality standards (losses). One advantage of advancing efforts in bioeconomy research is that the proportion of current production with zero economic value could be reduced via "biowaste valorization," diverting what would otherwise be losses into economically profitable inputs for other uses and transacted in other markets. To illustrate the link between the distribution of product quality subject to farmer control and the level of losses, we present evidence for the case of Chilean cherries, a potential source of antioxidant polyphenols and other compounds, and where a surprisingly high proportion of fruit is lost due to the failure to meet quality standards. Technical research elsewhere [30] shows that the notably large proportion of cherries unsaleable in traditional markets around the world will likely lead to the future introduction of alternative bioeconomy uses for the fruit.

However, accessing the bioeconomy alternative requires investments, modelled in this present paper, as some fix cost independent of productivity and quality decisions. With enough incentive to redirect some part of production from traditional markets to alternative markets, the distribution of product qualities shifts to lower average quality according to the standards of the traditional market. Our modelling framework allows an analysis of farmers' decisions regarding quality levels within a bioeconomy context and draws out the implications for the likelihood of farmer participation in new bioeconomy activities. We show that while the proportion of production with zero economic value (i.e., losses) decline, sales into traditional markets and product quality also decline. The approach taken here to understanding the incentives that alter a product's optimal quality distribution and loss levels and that simultaneously induce participation in an alternative market (where traditional quality is no longer important) contributes to linking the economic study of farm-level quality and food loss with the analysis of adoption decisions regarding bioeconomy activities. We illustrate the abstract, algebraic model with a numerical example to show the implications of altering economic incentives via the introduction of an alternative use for otherwise lost production. The results underline that with the bioeconomy alternative the opportunity cost of sales to traditional markets increases, leading to a decrease in sales to those markets, and leading to a shift downward in the distribution of product quality as demanded by traditional consumers. The introduction of an alternative market opportunity at worst leaves the farmer indifferent but would enhance the farmer's welfare, if fixed costs of accessing the new bioeconomy are sufficiently low. Nevertheless, a bioeconomy alternative would induce a reduction in average agricultural product quality and sales into traditional markets. Both low-productivity farms and high productivity farms are less likely to enter into a new market, while intermediate-level productivity farms are the most likely to adopt and shift to lower quality levels in response to alternative market availability. One threat worth underlining here is the competition between alternative uses associated with the growing bioeconomy and traditional uses of bio-based raw materials, especially food production. To the extent that traditional food, especially internationally traded staples, would tend to derive from relatively productive and mid- to large-scale farms in the developed world of middle- and high-income countries, the growth of bio-based alternative outlets could result in price competition, reduced supplies and increases in prices of food.

The analytical focus in this paper is on micro-level farmer decisions, and future empirical applications at the level of farmer production strategies related to quality and losses are required to give quantitative measures for specific cases. The authors are currently undertaking a study of Chilean cherry growers' production strategies related to quality and losses and their responses to changes in product prices across the range of traditional quality categories. In addition, and more generally, further research into aggregate supply

and demand consequences will be required in order to clarify the real-world implications of bioeconomy research and policies for various agricultural markets in the aggregate. The policy implications of the analysis, however, should elicit some additional concern among researchers and analysts interested in advancing the bioeconomy, given the potential negative impacts on food buyers and other final consumers with the almost-inevitable future introduction of bioeconomy alternatives for agricultural production.

**Author Contributions:** Conceptualization, S.J., W.F., and G.A.; methodology, S.J. and W.F.; software, S.J. and J.O.; validation, S.J. and W.F.; formal analysis, S.J. and W.F.; investigation, S.J.; writing—original draft preparation, S.J. and W.F.; writing—review and editing, S.J., W.F., G.A., and J.O.; visualization, W.F. and G.A.; supervision, S.J. All authors have read and agreed to the published version of the manuscript.

**Funding:** This research was funded by the National Agency for Research and Development (ANID)/ Scholarship Program/Doctorado Nacional/2018—21180109. Additional support was provided by Projet de Cooperation Bilaterale Wallonie-Bruxelles/Chile, WBI/AGCI, and by FAO's "Food losses and waste reduction and value chain development for food security in Egypt and Tunisia," funded by the AICS.

**Institutional Review Board Statement:** Not applicable.

**Informed Consent Statement:** Not applicable.

**Data Availability Statement:** Restrictions apply to the availability of these data. Commercial data was obtained from Nature South Exports, fruit exporter. Partial data is available from corresponding author upon request with the permission of the exporting firm.

**Acknowledgments:** We would like to thank Andrés Ruiz for his kind assistance with data and for being so generous with his expert knowledge in all aspects of cherry quality and marketing.

**Conflicts of Interest:** The authors declare no conflict of interest.

## Appendix A

The following lays out the details of the numerical examples presented in the text. The farmer's problem would be to maximize profit ($\pi$) by choosing to enter into a new market, d, or not (0 or 1) and the proportion of food that can be sold to the traditional and alternative markets:

$$\pi(P_t, P_b, f, y) = \max_{d,\theta}\{P_t\theta y + dP_b(1-\theta)y - C(\theta) - fd\} \tag{A1}$$

In the first scenario (not participate in the bioeconomy activity), the farmer's conditional maximization problem is:

$$\pi_0 = \max_{\theta} P_t\theta y - C(\theta) \tag{A2}$$

In the second scenario (participation), the farmer's profit function given the food prices can be expressed as:

$$\pi_1 = \max_{\theta} P_t\theta y + P_b(1-\theta)y - C(\theta) - f \tag{A3}$$

Given the quadratic cost function $C(\theta) = \frac{k}{2}\theta^2$ and evaluating the optimal level of food for $\theta_0^*$ in the first scenario, profits are:

$$\pi_0{}^* = \frac{P_t{}^2 y^2}{k} - \frac{k}{2}\frac{P_t{}^2 y^2}{k^2} \tag{A4}$$

$$\pi_0{}^* = \frac{1}{2}\frac{P_t{}^2 y^2}{k}$$

Replacing the optimal level of $\theta_1^*$ for the second scenario, profits are expressed as:

$$\pi_1{}^* = \frac{1}{2}\left[\frac{(P_t - P_b)^2 y^2}{k}\right] + P_b y - f \tag{A5}$$

The farmer enters the bioeconomy, if the benefits of doing so exceed the opportunity cost: $\pi_1^* \geq \pi_0^*$. Replacing $\pi_0^*$ and $\pi_1^*$ in the inequality:

$$\frac{y^2}{2k}\left[(P_t - P_b)^2 - P_t{}^2\right] + P_b y - f \geq 0 \tag{A6}$$

At a threshold level of fixed costs, $f^*$, the farmer would be indifferent between entering or not into a new market

$$\frac{y^2}{2k}\left[(P_t - P_b)^2 - P_t{}^2\right] + P_b y = f^* \tag{A7}$$

If there is no fixed cost associated with entering into a new market, $f = 0$, and given the productivity level ($y$), it is possible to set a value of the cost-scaling parameter $k$ which allows the farmer to be indifferent to both scenarios. More interestingly with respect to the likely heterogeneity of farmers is the relationship between the decision to participate in the alternative activity ($d$) and the productivity level ($y$), both then determining the level of sales in traditional and alternative markets and the level of losses. There are threshold levels of productivity where the farmer is indifferent between entering into a new market or not. Thus, from Equation (A7) a quadratic equation expressing this indifference can be written as:

$$y^2\left[(P_t - P_b)^2 - P_t{}^2\right] + 2kP_b y - 2fk = 0 \tag{A8}$$

The two roots of the quadratic equation (The determinant ($\Delta$) of Equation (A8) is $= 4k^2 P_b^2 - 16 P_t P_b fk + 8 P_b^2 fk$. It should be noted that if the discriminant is zero, there is a single root. If the discriminant is positive, there are two different roots for $h$. However, if the discriminant is negative, there is no real root for $y$.) (and thus the indifference limits for participating in the alternative activity) are:

$$y_1 = \frac{-2kP_b + \sqrt{4k^2 P_b^2 - 16 P_t P_b fk + 8 P_b^2 fk}}{2\left(-2P_t P_b + P_b^2\right)} \tag{A9}$$

$$y_2 = \frac{-2kP_b - \sqrt{4k^2 P_b^2 - 16 P_t P_b fk + 8 P_b^2 fk}}{2\left(-2P_t P_b + P_b^2\right)}$$

To maintain realism, the optimal proportion marketed, $\theta^*$, must range between zero and 1. To keep the algebra simple, to satisfy Equation (7) in the main text and avoid an uninteresting corner solution at $\theta^* = 1$, we restrict the interval for $y$ to the following range:

$$0 < y \leq \frac{k}{P_t} \leq \frac{k}{P_t - P_b} \tag{A10}$$

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
