# Peer review of "Understanding Farm-Level Incentives within the Bioeconomy Framework: Prices, Product Quality, Losses, and Bio-Based Alternatives"

_sustainability, doi:10.3390/su13020450_

Round 1

Reviewer 1 Report

The manuscript addresses the important topic of commercialising off-spec food products by the example of off-spec cherries. Specifically the manuscript analysis the decision steps a farmer has to take.

The model is described in great detail. However, the practical relevance of the model suffers from the fact that the example of the cherries is not explained in detail. It would be interesting to demonstrate concretely at what price of a certain cherry quality it is advantageous to market antioxidants at a certain price.
Overall, the text is very detailed and I recommend shortening it considerably.

Reviewer 2 Report

The paper attempts to answer the question how the shift towards bioeconomy will affect farmers' decisions concerning production (quantity, quality, markets). The Authors use a microeconomic model of profit maximization and show the results for farmers' decisions when an alternative market appears. The paper contributes to understanding firm-level processes related to the bioeconomy development. However, I'd suggest some improvements.

When discussing the decision framework (ln.110 onwards) the Authors don't mention uncertainty and risk related to the agricultural production, i.e. the influence of external conditions (especially weather events) on the prodcution size and quality. I understand why this was not included in the model but I suggest mentioning this in the literature review.

The introduction also lacks an oerview of methodological approaches. What methods are applied for similar analyses? Why this particular method was applied? 

The research design is based on a standard microeconomic optimisation model which was used to show that the emergence of an alternative market will cause producers to direct some of their production to it and that this effect depends on the price relationship between the traditional and alternative markets and the costs of entering the alternative market. I appreciate the formal aspect of the model, but this effect is quite obvious. The added value of the paper should be more underlined here. Have the Authors considered the empirical application of the model?

I'd say that the Authors stop their deliberations at the point where the conclusions start to be most interesting: They find that the farmers with a mid-level productivity have the most incentives to enter the alternative  market (ln.470-480). What are practical implications of this observation? What are consequences for the public policy?

In my opinion, the conclusions concerning public policies could be discussed more widely. The main threat here is the competition between tranditional uses of bio-based raw materials, especially food production, and alternative uses. What if price competition between these two markets will result in reduced supplies and increase in prices of food? This is mentioned, e.g. ln. 288-292, but deserves more attention.

The last comment is purely technical. The Figure 2 seems to be repeated three times.

Reviewer 3 Report

Dear Authors, I like your article so i am sympathetic with the final judgment. However, some changes are necessary to improve quality of your paper.

1) line 255, strange Spanish

2) Title. Diversion, Land-use. Strange terms, please consider land use conversion or land use change

3) sustainability is a non-modelist journal so I would encourage to enrich the substantive part and keep the model for the appendix.

4) you have to clarify better the importance and scope of the numerical exercise.

5) you have to explain if the model works enough with field data (farms).

6) Is the model general enough and working at what scale? Farms?

7) Probably a better explaination of scope and relevance of the paper will be necessary.

8) Bibliography should recognize general problems associated with the issues detailed in the title. Some of them were not clearly addressed in the text (e.g. land-use change).

Thank you.

Round 2

Reviewer 1 Report

This manuscript analyzes the additional options that bioeconomic utilization of biomass offers to producers. As an example, the utilization of cherries in Chile is described. It is convincingly shown that the utilization of off-spec fruits can reduce the amount of biomass waste and increase the income of producers. Analyzing this practical example theoretically is a contribution to the scientific discussion of the bioeconomy.
The layout of the manuscript needs to be revised. Fig. 2 is visible in triplicate; spelling errors are in line 371, for example.

Author Response

The layout of the manuscript needs to be revised. Fig. 2 is visible in triplicate; spelling errors are in line 371, for example.

Response: We did not see any obvious spelling errors on line 371. We did note on that line that we had referred to a figure without numeration. We now referred explicitly to figure 3.

With respect to the triplicate of Fig. 2, we do not observe the multiple figures and we do not know what is going on. We will communicate with the editor to make sure that this problem will be resolved.

We would like thank the reviewer for the comments during the first round of revisions, which improved our final paper. 

Reviewer 2 Report

The paper presents a theoretical model of the effects of bioeconomy development. Considerations are based on a standard, deterministic profit maximization model. The novelty of the conclusions lies in showing a microeconomic mechanism and potential threats of bioeconomy development.

There are some minor errors in the text: figure 2 appears three times, figure references should also be corrected (ln.371).

I understand that the newly added part of the introduction (lines 120-138) presents the hypotheses of the paper, but this could be stressed more. 

Author Response

1: There are some minor errors in the text: figure 2 appears three times, figure references should also be corrected (ln.371)

Response 1: We now referred explicitly to figure 3.

With respect to the triplicate of Fig. 2, we do not observe the multiple figures and we do not know what is going on. We will communicate with the editor to make sure that this problem will be resolved.

2: I understand that the newly added part of the introduction (lines 120-138) presents the hypotheses of the paper, but this could be stressed more. 

Response 2:  In order to stress the contribution of our approach and the hypotheses of this paper, we have added to the conclusion section a new paragraph between 625 and 637 lines.

We would like to thank the reviewer for the comments and suggestions during the review process, which have contributed to improving our paper considerably.  

Reviewer 3 Report

Good revision overall. The article is interesting and can be published as it is. thank you.

Author Response

We thank the reviewer for the comments which improved the revision.